⊘ | **Open Peer Review** | Clinical Microbiology | Research Article

# Evaluation of the Bruker MALDI Biotyper (MBT) subtyping module for detection of *Klebsiella pneumoniae* carbapenemase (KPC) in *Enterobacterales* in a Canadian clinical microbiology laboratory

Emma Finlayson-Trick,[1,2] Heather Glassman,[1,2] Jasmine Ahmed-Bentley,[1,2] Xinhe Liu,[1] Linda Tsui,[1] Lori Sung,[1] Charlene Porter,[1] Claudine Desruisseaux,[1,2] Valery Lavergne,[1,2] Suefay Liu,[1,2] Jennifer Tat,[1,2] Anthony Lieu,[1,2,3] Aruna Uma Chandran,[1] Marthe K. Charles[1,2]

**ABSTRACT** Early detection of *Klebsiella pneumoniae* carbapenemase (KPC)-producing *Enterobacterales* enables prompt initiation of appropriate antimicrobial therapy and infection prevention and control measures. The matrix-assisted laser desorption/ionization time-of-flight (MALDI-ToF) Biotyper (MBT) Subtyping Module (Bruker Daltonics) uses a characteristic spectral peak at ~11,109 m/z to identify KPC-producing *Enterobacterales*. The aim of this study was to assess the ability of the MBT Module to detect KPC among a variety of clinical and reference (AR Bank) *Enterobacterales* isolates. Detection of KPC was compared between the MBT Module and a laboratory-developed real-time KPC PCR. Of the 93 isolates tested, 58 (62.4%) were positive for KPC by PCR (39 clinical and 19 AR bank isolates), and 35 were negative. Among the KPC-positive PCR isolates, nine (seven clinical and two AR bank) also tested positive for KPC by the MBT Module. KPC detection via the MBT Module had a sensitivity of 15.5% and a specificity and positive predictive value of 100%. This study also assessed the performance of the MBT Module to detect KPC-2 versus KPC-3 among the AR Bank isolates. The MBT Module detected 2/5 (40.0%) KPC-2 and 0/14 of KPC-3 AR Bank isolates. As sensitivity was low, but specificity was high, a positive MBT Module result reliably predicts the presence of KPC; however, negative results cannot be relied upon to be truly KPC-negative.

**IMPORTANCE** *Enterobacterales*, a large order of pathogenic and commensal bacteria, can gain resistance to carbapenems through the acquisition of plasmids carrying genes, such as *Klebsiella pneumoniae* carbapenemase (KPC). Early detection of KPC is crucial for patient care; however, current detection methods that involve multiple time-consuming steps, commonly only undertaken after overnight susceptibility testing, indicate possible carbapenem resistance. This study examined the ability of matrix-assisted laser desorption/ionization time-of-flight (MALDI-ToF), a common diagnostic tool in the clinical laboratory, to detect KPC among *Enterobacterales*. KPC detection via MALDI-ToF had low sensitivity but 100% specificity. These results support and expand the microbiological and geographical range of prior publications for this tool. A practical application of this highly specific method would involve rapid triaging of KPC-positive isolates a day earlier than current methods, permitting initiation of more timely therapeutic and infection prevention and control measures.

**KEYWORDS** Carbapenemase, KPC, MALDI-ToF

**Peer Reviewers** Marcelo Pillonetto, Pontifícia Universidade Catolica do Parana Escola de Medicina e Ciencias da Vida, Paraná, Brazil; Richard D. Smith, University of Maryland School of Medicine, Baltimore, Maryland, USA

Address correspondence to Marthe K. Charles, marthe.charles@vch.ca.

The authors declare no conflict of interest.

Carbapenemase-producing *Enterobacterales* (CPE) present a significant threat to global health (1). Located on mobile genetic elements, such as plasmids, carbapenemases are easily transferred among bacteria, facilitating the spread of multi-drug resistance (2). Historically in Canada, CPE infections were attributed to international travel, but in recent years, an increasing number of cases have been associated with domestic acquisition (3–5). *Klebsiella pneumoniae* carbapenemase (KPC), New Delhi metallo-β-lactamase (NDM), and oxacillinase 48 (OXA-48) account for 86–96% of carbapenemases in Canada (6). Nationally, KPC is implicated in more than half of all CPE infections (6). In British Columbia, while the total number of KPC cases is relatively low, the case count has been steadily increasing over the last decade (Fig. 1) (7, 8).

In Canada, most clinical microbiology laboratories routinely use matrix-assisted laser desorption/ionization time-of-flight (MALDI-ToF), a mass spectrometry-based method, to rapidly identify organisms with high sensitivity and specificity. Recent studies have demonstrated that in addition to identification, MALDI-ToF can be leveraged to detect antibiotic resistance genes, including carbapenemases like KPC (9–11). This approach has the potential to significantly shorten time to KPC detection, as current investigations, such as the NG-Test CARBA 5 (Hardy Diagnostics, Santa Maria, CA) or CPE PCR, are only initiated after the bacteria have been identified, and the susceptibility pattern has been confirmed (Fig. 2). The MALDI Biotyper (MBT) Subtyping Module (MBT Module, Bruker Daltonics) uses a characteristic spectral peak at ~11,109 m/z to identify KPC among *Citrobacter freundii*, *Enterobacter aerogenes*, *Enterobacter asburiae*, *Enterobacter cloacae*, *Enterobacter kobei*, *Enterobacter ludwigii*, *Escherichia coli*, *Klebsiella aerogenes*, *Klebsiella oxytoca*, *Klebsiella pneumoniae*, *Klebsiella variicola*, and *Serratia marcescens* (12). This peak corresponds to a protein encoded by the *p019* gene located adjacent to $bla_{KPC}$, which is associated with 39% of KPC-2 isolates and 4% of KPC-3 isolates (13). As such, studies indicate that KPC detection by MALDI-ToF has high specificity (99%) but low sensitivity (42–85%) (14, 15).

It is crucial that clinical microbiology laboratories quickly and accurately detect KPC-producing *Enterobacterales* so that physicians can initiate appropriate broad-spectrum empiric antibiotics, such as meropenem-vaborbactam, ceftazidime-avibactam, imipenem-cilastatin-relebactam, or cefiderocol, which is necessary for patient survival (16, 17). Moreover, early detection of KPC enables implementation of appropriate infection prevention and control (IPAC) precautions, which is key for controlling the spread of KPC in hospital or long-term care facilities (18). Currently, few publications have used the MBT Module to detect KPC beyond *Klebsiella* species (9–11, 13–15, 19, 20). Moreover, none of the publications in this field have involved Canadian clinical

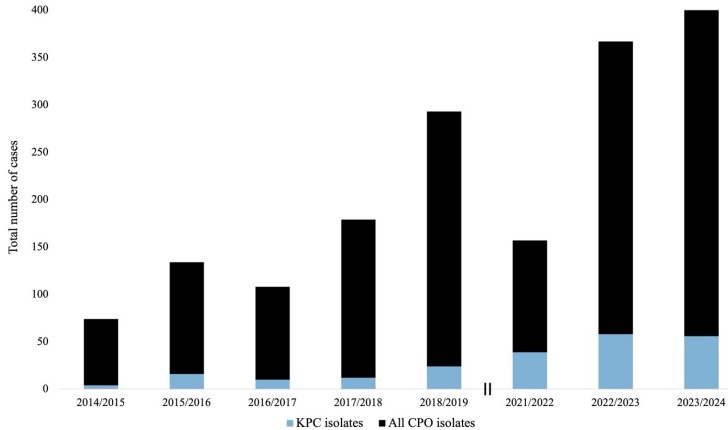

**FIG 1** Annual number of cases due to KPC-producing organisms (blue) compared to number of cases due to all carbapenemase-producing organisms (CPO, black) in British Columbia, Canada. The double vertical line indicates a gap in surveillance data between 2019–2021. The figure was created using surveillance data collected from the Provincial Infection Control Network of British Columbia (8).

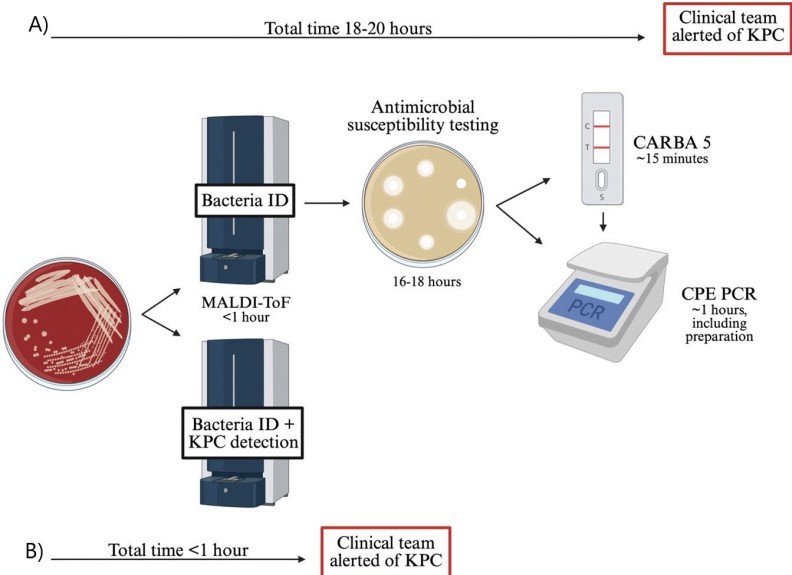

**FIG 2** Comparison between the current workflow and the MBT Module for KPC detection. Following subculture, bacterial species are identified via MALDI-ToF. In the current workflow (A), KPC detection and clinical notification occur only after susceptibility testing and specific carbapenemase testing with CARBA 5 and/or CPE PCR. The MBT Module (B) enables earlier detection of KPC at the time of bacterial identification, allowing for timely clinical notification (about a day earlier than the current pathway).

isolates. This study involved validating the MBT Module for KPC detection in a Canadian hospital laboratory using a variety of *Enterobacterales* isolates from clinical samples and the Antimicrobial Resistance Isolate Bank (AR Bank) (21).

## MATERIALS AND METHODS

### Preparation of isolates for testing

A data set of *Enterobacterales* isolates was selected to reflect the historic trends of KPC in the hospital clinical laboratory and was based on prior phenotypic and molecular results that demonstrated either the presence or absence of KPC (Table 1). This frozen data set included 70 clinical isolates (both patient specimen and surveillance samples) from a hospital laboratory. Clinical isolates were limited to one per patient and were selected between May 2022 and August 2024. The remainder of the data set included 23 isolates from the AR Bank (gram-negative carbapenemase detection [CarbaNP] and *Enterobacterales* carbapenemase diversity [CRE] panels). Isolates were subcultured twice on 5% sheep blood agar prior to testing.

### MALDI-ToF MS testing

Culture plates with 24 ± 8 h of growth were accepted for MALDI-ToF MS testing. MALDI-ToF MS was performed as per the manufacturer's instructions. All isolates were tested on both the MALDI Biotyper Sirius and Smart System (Bruker Daltonics, Biller-ica, MA). The mass spectral readouts were obtained using MBT HT Client (Version 5.1.410.9032). The detailed report was reviewed to confirm isolates that were identified as KPC-positive. As per the manufacturer's guidelines, KPC subtyping was not completed on isolates with low identification scores. Therefore, any isolate with an ID score < 2.0 were re-spotted and re-tested the same day.

**TABLE 1** Species distribution of clinical and AR Bank samples KPC-producing and non-KPC-producing bacterial isolates (total $N = 93$)[a]

| Species | Total isolates | KPC-producing isolates | Non-KPC producing isolates |
|---|---|---|---|
| Clinical samples | | | |
| *K. pneumoniae* complex | 24 | 15 | 9 |
| *C. freundii* | 17 | 12 | 5 |
| *E. coli* | 11 | 4 | 7 |
| *K. oxytoca* | 7 | 6 | 1 |
| *E. cloacae complex* | 6 | 1 | 5 |
| *S. marcescens* | 2 | 0 | 2 |
| *E. bugandensis* | 1 | 1 | 0 |
| *K. aerogenes* | 1 | 0 | 1 |
| AR Bank samples | | | |
| *K. pneumoniae* complex | 13 | 11 | 2 |
| *E. coli* | 5 | 2 | 3 |
| *E. cloacae complex* | 4 | 4 | 0 |
| *C. freundii* | 1 | 1 | 0 |
| *K. oxytoca* | 1 | 1 | 0 |

[a]AR Bank isolate numbers are as follows: *K. pneumoniae* (0097, 0098, 0112, 0113, 0115, 0117, 0120, 0125, 0126, 129, 041, 068), *E. coli* (0061, 0114, 048, 0150, 0162), *E. cloacae* (0032, 0053, 0136, 0163), *C. freundii* (0116), *K. oxytoca* (0147), and *K. ozaenae* (0096).

## KPC PCR

This multiplex real-time colony PCR was used as the gold standard to confirm the presence or absence of KPC in the tested isolates. This PCR was developed by the British Columbia Center for Disease Control Public Health Laboratory (22). Primers and a fluorescent probe (outlined in Table 2) were obtained from Integrated DNA Technologies (Coralville, IA).

Approximately one to three bacterial colonies from each culture plate were inoculated into tubes with sterile water (turbidity approximately 0.5 McFarland standard). The tubes were then heated at 100°C for 8 min to promote bacterial lysis.

As per the manufacturer's instructions, each PCR reaction contained TaqMan Fast Advanced Master Mix (Life Technologies, Burlington, ON), RNase-free water, primers and probe, and sample lysate. Real-time PCR was conducted on the ABI 7500 platform (ThermoFisher, Waltham, MA) with the recommended fast thermal cycling conditions (35 cycles total). The PCR reaction was considered valid if the positive control cycle threshold (Ct) value was between 20 and 30, and all the negative controls were negative for KPC. An isolate was considered positive for KPC if the Ct value was between 10 and 30.

### Analysis and preparation of figures

Performance was evaluated by comparing the MBT Module against a real-time PCR assay as the reference standard. Sensitivity was calculated as the percentage of the isolates detected as KPC-positive by the MBT Module among those positive by KPC PCR. Specificity was calculated as the percentage of the isolates detected as KPC-negative by the MBT Module among those negative by KPC PCR. Positive predictive value (PPV) was determined by calculating the proportion of isolates truly carrying KPC (true positives) by the KPC-positive isolates detected via the MBT Module (total positive tests). Negative predictive value (NPV) was determined by calculating the proportion of isolates lacking

**TABLE 2** Primers and probe for KPC detection

| Name | Sequence (5′ → 3′) | Reporter/quencher | [Final] (µM) |
|---|---|---|---|
| KPC Forward | GGCCGCCGTGCAATAC | | 0.2 |
| KPC Reverse | GCCGCCCAACTCCTTCA | | 0.2 |
| KPC Probe | TGATAACGCCGCCGCCAATTTGT | MAX/ZEN/IFBQ | 0.15 |

KPC (true negatives) by the total number of negative tests (true and false negatives). The figure was created in BioRender (2024).

## RESULTS

In total, 93 isolates (clinical and AR Bank) were tested, of which 58 (62.4%) were confirmed to carry a gene for KPC via PCR (Table 3). Of those PCR-positive isolates, nine (seven clinical, two AR Bank) were identified as KPC-positive via the MBT Module. There were no discordant results between the Sirius and Smart systems. Overall sensitivity was 15.5% (9/58), and specificity was 100%. There were no instances where the MBT Module result was positive, but PCR was negative. As such, the PPV of the MBT module was 100%. The NPV of the MBT module was 41.7% (35 isolates were true negatives; 49 isolates were false negatives).

KPC types were only available for the AR Bank isolates of which five were KPC-2 and 14 were KPC-3. The MBT Module identified two AR Bank isolates as KPC positive, both of which were KPC-2. As such, the frequency of positive KPC identification was 40% in the KPC-2 group and 0% in the KPC-3 group.

## DISCUSSION

In this study, the MBT Module had a low sensitivity (15.5%) but high specificity (100%) and PPV (100%) for the detection of KPC-producing *Enterobacterales*. These results support similar findings (42% sensitivity, 100% specificity) published by Cuello et al. (13) who tested more than 700 clinical *Klebsiella* isolates from the United States and South America (13). This study expanded upon the work of Cuello et al. (13) by contributing geographic representation from Canada and by analyzing the performance of the MBT Module for KPC detection among *Klebsiella* and non-*Klebsiella* bacterial isolates.

In addition to the methods used in this study, there are other automated approaches for rapid KPC detection in *Enterobacterales*. A study by Centonze et al. (19) similarly targeted the ~11,109 m/z peak but used the MALDI-ToF Vitek MS system from Bio-Mérieux (19). In their sample of 176 KPC-positive clinical isolates, they reported high sensitivity (99.4%) and specificity (100%) for KPC detection using the MALDI-ToF Vitek MS System (19). The authors noted, however, that the high sensitivity and specificity may be a result of testing isolates derived from the same clone (19). Other groups have also improved the sensitivity of detection by targeting the mature KPC protein (~28,643–28,731 Da) rather than the biomarker (~11,109 Da). For example, Costa et al. (14) reported a sensitivity and specificity of 100% for the detection of the mature KPC protein in their data set of clinical blood cultures, short-term cultures, and colonies, but sensitivity dropped to 63% when the biomarker was targeted (14). Similarly, Moreira et al. (20) reported a sensitivity and specificity of 98.1 and 97.9%, respectively, when they targeted the mature KPC protein in their sample of clinical *Enterobacterales* isolates (20). Comparing detection of the mature protein with the p019 biomarker was not part of this study; meanwhile, it could represent a future research direction. As such, different approaches can be taken for KPC detection depending on the availability of equipment in the clinical laboratory.

TABLE 3 Comparison of KPC-positive isolates (total $N = 58$) detected via PCR ($N = 58$) and MALDI-ToF (total $n = 9$)

| | KPC detected via PCR/total number of isolates | KPC detected via MBT module | Sensitivity (%) |
|---|---|---|---|
| *K. pneumoniae* complex | 26/37 | 2 | 7.7 |
| *C. freundii* | 13/18 | 2 | 15.4 |
| *K. oxytoca* | 7/8 | 5 | 71.4 |
| *E. coli* | 6/16 | 0 | 0 |
| *E. cloacae* complex | 5/10 | 0 | 0 |
| *E. bugandensis* | 1/1 | 0 | 0 |

This validation examined the ability of the MBT Module to detect KPC among a variety of *Enterobacterales*. The distribution of *Enterobacterales* species was selected to reflect the isolates previously observed in the hospital's clinical laboratory between 2022 and 2023, whereby among 42 KPC-producing *Enterobacterales*, 19 were *K. pneumoniae* (45 and 39% in the present study), and six were *C. freundii* complex (14 and 19% in the present study). In the literature, the presence of *p019* is more commonly detected in *Klebsiella* species compared to *Citrobacter* species, although a far fewer number of *Citrobacter* isolates have been studied (10, 23, 24). Similar to previous publications, the MBT Module detected KPC in some of the *K. oxytoca*, *C. freundii*, and *K. pneumoniae* isolates. As such, the composition of *Enterobacterales* species used in this study may have contributed to the low MBT Module sensitivity. Furthermore, unlike Centonze et al. (19) who detected 92.3% (12/13) of KPC-producing *E. coli* in their data set, the MBT Module in this study did not detect any KPC-producing *E. coli* isolates (19). It is possible that the KPC-producing *E. coli* isolates in this study differ significantly in their carriage of *p019* due to geographic variation. In a study of KPC-positive *K. pneumoniae* isolates across Europe, Gato et al. (25) observed wide geographic variation of *p019* detection from 100% in France and Ireland to 0% in Portugal and Poland (25). As *p019* is part of a mobile genetic element, it is feasible that through selective pressure, many of this study's isolates maintained the resistance gene (*blaKPC*) but lost *p019* (25). Consequently, the low MBT Module sensitivity seen here may indirectly reflect a local plasmid reservoir that lacks *p019*. Otherwise, the MBT Module did not detect KPC in *E. bugandensis* and *K. ozaenae*, but $bla_{KPC}$ is not automatically detected in these bacteria as noted in the Bruker manual. While the sample size was sufficient for validation in the clinical laboratory, the small number of isolates was a limitation that meant that some bacterial species were rarely tested. As such, these findings require external evaluation for confirmation.

Future studies are needed to better understand the characteristics of KPC detection by the MBT Module. In the laboratory, it remains unclear if there are other factors, besides the presence/absence of *p019*, influencing KPC detection among the various *Enterobacterales* species. Furthermore, additional work is needed to evaluate how KPC detection by the MBT Module impacts quality indicators in the microbiology laboratory and patient care. While this study combined surveillance and patient specimens, it would be interesting to tease apart these sample types when evaluating impact given the different IPAC implications. Finally, this work with KPC invites the opportunity to explore more targets for the MBT Module, including other carbapenemases like NDM and OXA-48 (26, 27).

In conclusion, despite a low sensitivity, the automated MBT Module had high specificity and PPV for the detection of KPC-producing *Enterobacterales*, making it a potentially useful adjunctive "rule-in" tool at the time of MALDI-ToF identification, prior to the availability of conventional susceptibility testing (10, 28, 29). Many clinical laboratories already use MALDI-ToF for bacterial identification; therefore, detection of KPC by the MBT Module would not significantly change the current workflow and would not require the purchase of an additional instrument. It is important to note that due to the low sensitivity, all negative MBT Module results would still require routine testing to evaluate for KPC, as would positive results to evaluate for the presence of additional CPE genes like NDM or OXA-48. Nevertheless, KPC detection at the time of bacterial identification has the potential to critically impact patient care by facilitating an earlier switch to more appropriate antibiotics and ensuring suitable IPAC precautions.

## ACKNOWLEDGMENTS

We would like to thank the medical laboratory technologists at Vancouver General Hospital for their contribution to this project.

## AUTHOR AFFILIATIONS

[1]Division of Medical Microbiology & Infection Control, Vancouver Coastal Health Authority, Vancouver, British Columbia, Canada

2Department of Pathology and Laboratory Medicine, University of British Columbia, Vancouver, British Columbia, Canada

3Deparment of Medicine, Division of Infectious Diseases, University of British Columbia, Vancouver, British Columbia, Canada

## AUTHOR ORCIDs

Emma Finlayson-Trick ⓘ http://orcid.org/0000-0002-3287-041X

Jasmine Ahmed-Bentley ⓘ http://orcid.org/0009-0003-1714-5548

Marthe K. Charles ⓘ http://orcid.org/0000-0002-3663-0974

## AUTHOR CONTRIBUTIONS

Emma Finlayson-Trick, Formal analysis, Visualization, Writing – original draft, Writing – review and editing | Heather Glassman, Conceptualization, Data curation, Formal analysis, Investigation, Methodology, Validation, Writing – review and editing | Jasmine Ahmed-Bentley, Writing – review and editing | Linda Tsui, Investigation | Lori Sung, Investigation | Charlene Porter, Investigation | Claudine Desruisseaux, Investigation, Writing – review and editing | Valery Lavergne, Writing – review and editing | Suefay Liu, Writing – review and editing | Jennifer Tat, Writing – review and editing | Anthony Lieu, Writing – review and editing | Marthe K. Charles, Conceptualization, Formal analysis, Funding acquisition, Methodology, Project administration, Supervision, Writing – review and editing.

## ADDITIONAL FILES

The following material is available online.

Open Peer Review

**PEER REVIEW HISTORY (review-history.pdf).** An accounting of the reviewer comments and feedback.

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
