## [Reviewer comments · Microbiology Spectrum]

Microbiology Spectrum

Evaluation of the Bruker MALDI Biotyper® (MBT) Subtyping Module for detection of *Klebsiella pneumoniae* Carbapenemase (KPC) in Enterobacterales in a Canadian clinical microbiology laboratory

Emma Finlayson-Trick, Heather Glassman, Jasmine Ahmed-Bentley, Xinhe Liu, Linda Tsui, Lori Sung, Charlene Porter, Claudine Desruisseaux, Valery Lavergne, Suefay Liu, Jennifer Tat, Anthony Lieu, Aruna Chandran, and Marthe Charles

Corresponding Author(s): Marthe Charles, Vancouver Coastal Health Authority

Review Timeline:

Submission Date:	August 29, 2025
Editorial Decision:	October 1, 2025
Revision Received:	November 19, 2025
Accepted:	November 21, 2025

Editor: Paul Luethy

Reviewer(s): Disclosure of reviewer identity is with reference to reviewer comments included in decision letter(s). The following individuals involved in review of your submission have agreed to reveal their identity: Marcelo Pilonetto (Reviewer #1); Richard D. Smith (Reviewer #2)

Transaction Report:

DOI: <https://doi.org/10.1128/spectrum.02511-25>

Re: Spectrum02511-25 (**Evaluation of the Bruker MALDI Biotyper® (MBT) Subtyping Module for detection of *Klebsiella pneumoniae* Carbapenemase (KPC) in Enterobacterales in a Canadian clinical microbiology laboratory**)

Dear Dr. Marthe Charles:

Thank you for the privilege of reviewing your work. Below you will find my comments, instructions from the Spectrum editorial office, and the reviewer comments.

Revision Guidelines

Sincerely,
Paul Luethy
Editor
Microbiology Spectrum

Reviewer #1 (Comments for the Author):

The manuscript presents a comprehensive validation of the Bruker MALDI Biotyper Subtyping Module (MBT Module) for the rapid detection of *Klebsiella pneumoniae* Carbapenemase (KPC) in a cohort of 93 Enterobacterales isolates, comprising both Canadian clinical samples and reference isolates from the Antimicrobial Resistance (AR) Bank. This study, utilizing real-time KPC PCR as the gold standard, yielded results that confirm both the potential utility and significant limitations of the commercial

assay in a distinct epidemiological setting. The core quantitative findings demonstrate a dichotomy in assay performance: of the 58 isolates confirmed KPC-positive by PCR, the MBT Module successfully detected only nine. This resulted in an overall sensitivity of 15.5% (9/58). However, the module achieved 100% specificity and a 100% positive predictive value (PPV), as there were zero instances where the MBT Module reported a positive result that was contradicted by the reference PCR.

Suggestions:

1- Material and Methods section:

Your results show a very low sensitivity (15.5%), which is substantially lower than previously published metrics, which generally report sensitivities ranging from 42% to 85%.

Was this dataset selected randomly? I believe the percentage of other Enterobacterales than *K. pneumoniae* could be potentially biasing such a low sensitivity.

I see specifically an overrepresentation of species like *Citrobacter* and an underrepresentation of *Klebsiella*. In this sense, I suggest including information that supports the notion that this species' distribution aligns with your routine findings. It seems the number of *C. freundii* KPC - positive is too high. I would recommend reproducing in the study the approximate percentage of isolates you find in your routine results. Otherwise, the test performance could be severely affected.

2- Results sections:

It would be very interesting to see the results in Table 3 present the performance of the proposed MALDI test for KPC-2 and KPC-3, separately, since the authors mention that the samples were selected based on previous sequence results.

3- Discussion section:

The authors should discuss further whether this low sensitivity is due to geographical validation or sample biases, as this study is the first to use Canadian clinical isolates for this evaluation.

The high prevalence of p019-negative KPC strains, inferred by the 15.5% sensitivity, suggests the local dominance of highly streamlined and potentially more promiscuous plasmids. Such plasmids, optimized for horizontal gene transfer, may selectively maintain the resistance gene (*bla*KPC), while jettisoning accessory elements like p019. Therefore, the low MBT sensitivity could function as an indirect indicator of a highly evolved, geographically adapted, and clinically resilient local plasmid reservoir. You should include this discussion in your manuscript.

4- Conclusion section:

The conclusion must unequivocally reiterate that 100% specificity and 100% PPV are the essential outcomes. This characteristic supports the MBT Module solely as a high-speed "rule-in" tool for initiating IPC and therapeutic triage earlier than current methods.

General suggestions:

It would be exciting and valuable to see the results for the mature KPC protein compared to the p019 protein in your dataset.

Reviewer #2 (Comments for the Author):

This study evaluates MBT module to identify KPC harboring organisms. While the data of this study is not promising, the negative data may have utility to educate other clinical laboratorians to steer away from using this assay. Overall, it is a sound validation but could use some improvements in displaying data. Furthermore, their conclusions do not match the data presented.

Major revisions

Line 68. Consider removing figure 1 as this is reused data and graphics from a different source

Line 99. Meropenem-vaborbactam is the drug of choice for KPC and the other drugs mentioned may not work if it is another carbapenemase. Consider revising since this study is only for KPC.

Line 104. Several studies have looked at KPC detection with MALDI. Please cite here.

Line 121. In table 1, consider organizing members of the *E. cloacae* complex and *K. pneumoniae* complex together as inaccuracies speciating can occur via MALDI.

Line 168. I would add negative predictive value to best illustrate impact from false negatives.

Line 172. Table 3 needs more detail of the results or consider deleting. To make table 3 better, I would add total number of isolates for each species, percent with KPC, sensitivity with numbers, and specificity without numbers.

Line 182. Although previous studies show poor results, why are the results of this study even worse? I would like to see a possible explanation.

Line 228. I disagree with the conclusions as an adjunctive tool. With 100% specificity all we know is that if it is right it is definitely right but if it is negative we should still be concerned. I do not think this would help stewardship efforts at all. What would be the

proposed implementation.

Line 228. I would like to see conclusions regarding the actual time saved with these poor results. Overall, I find the author's conclusions do not match what the data says.

Was any investigation into the presence of p019 in these isolates or variant of KPC? Phenotypic comparison would also be helpful via mCIM or Carba5.

**Evaluation of the Bruker MALDI Biotyper® (MBT) Subtyping Module for detection of**
***Klebsiella pneumoniae* Carbapenemase (KPC) in Enterobacterales in a Canadian clinical**
**microbiology laboratory**

Emma Finlayson-Trick^{1,2}, Heather Glassman^{1,2}, Jasmine Ahmed-Bentley^{1,2}, Xinhe Liu¹, Linda
Tsui¹, Lori Sung¹, Charlene Porter¹, Claudine Desruisseaux^{1,2}, Valery Lavergne^{1,2}, Suefay Liu^{1,2},
Jennifer Tat^{1,2}, Anthony Lieu¹⁻³, Aruna Uma Chandran¹, Marthe K. Charles^{1,2}

¹Division of Medical Microbiology & Infection Control, Vancouver Coastal Health Authority,
Vancouver, BC; ²Department of Pathology and Laboratory Medicine, University of British
Columbia, Vancouver, BC; ³Department of Medicine, Division of Infectious Diseases, University
of British Columbia

Corresponding author:

Dr. Marthe K. Charles

Email: marthe.charles@vch.ca

Key words: Carbapenemase, KPC, MALDI-ToF

Max word count: 5000

**Abstract** (250 words max, currently 214)

[revised manuscript text omitted]

- [23] Gato E, Arroyo MJ, Méndez G, Candela A, Rodiño-Janeiro BK, Fernández J, et al. Direct
Detection of Carbapenemase-Producing *Klebsiella pneumoniae* by MALDI-TOF Analysis of
Full Spectra Applying Machine Learning. *J Clin Microbiol* 2023;61:e01751-22.
<https://doi.org/10.1128/jcm.01751-22>.
- [24] Hrabák J, Študentová V, Walková R, Žemličková H, Jakubů V, Chudáčková E, et al.
Detection of NDM-1, VIM-1, KPC, OXA-48, and OXA-162 Carbapenemases by Matrix-
Assisted Laser Desorption Ionization–Time of Flight Mass Spectrometry. *J Clin Microbiol*
2012;50:2441–3. <https://doi.org/10.1128/jcm.01002-12>.
- [25] Cheon DH, Jang H, Choi YK, Oh WS, Hwang S, Park J-R, et al. Clinical evaluation of
advanced MALDI-TOF MS for carbapenemase subtyping in Gram-negative isolates. *J Clin*
*Microbiol* 2025;63. <https://doi.org/10.1128/jcm.01475-24>.

This study evaluates MBT module to identify KPC harboring organisms. While the data of this study is not promising, the negative data may have utility to educate other clinical laboratorians to steer away from using this assay. Overall, it is a sound validation but could use some improvements in displaying data. Furthermore, their conclusions do not match the data presented.

Major revisions

Line 68. Consider removing figure 1 as this is reused data and graphics from a different source

Line 99. Meropenem-vaborbactam is the drug of choice for KPC and the other drugs mentioned may not work if it is another carbapenemase. Consider revising since this study is only for KPC.

Line 104. Several studies have looked at KPC detection with MALDI. Please cite here.

Line 121. In table 1, consider organizing members of the *E. cloacae* complex and *K. pneumoniae* complex together as inaccuracies speciating can occur via MALDI.

Line 168. I would add negative predictive value to best illustrate impact from false negatives.

Line 172. Table 3 needs more detail of the results or consider deleting. To make table 3 better, I would add total number of isolates for each species, percent with KPC, sensitivity with numbers, and specificity without numbers.

Line 182. Although previous studies show poor results, why are the results of this study even worse? I would like to see a possible explanation.

Line 228. I disagree with the conclusions as an adjunctive tool. With 100% specificity all we know is that if it is right it is definitely right but if it is negative we should still be concerned. I do not think this would help stewardship efforts at all. What would be the proposed implementation.

Line 228. I would like to see conclusions regarding the actual time saved with these poor results. Overall, I find the author's conclusions do not match what the data says.

Was any investigation into the presence of p019 in these isolates or variant of KPC? Phenotypic comparison would also be helpful via mCIM or Carba5.

Minor revisions

Response to Reviewers

Reviewer #1 (Comments for the Author):

The manuscript presents a comprehensive validation of the Bruker MALDI Biotyper Subtyping Module (MBT Module) for the rapid detection of *Klebsiella pneumoniae* Carbapenemase (KPC) in a cohort of 93 Enterobacterales isolates, comprising both Canadian clinical samples and reference isolates from the Antimicrobial Resistance (AR) Bank. This study, utilizing real-time KPC PCR as the gold standard, yielded results that confirm both the potential utility and significant limitations of the commercial assay in a distinct epidemiological setting. The core quantitative findings demonstrate a dichotomy in assay performance: of the 58 isolates confirmed KPC-positive by PCR, the MBT Module successfully detected only nine. This resulted in an overall sensitivity of 15.5% (9/58). However, the module achieved 100% specificity and a 100% positive predictive value (PPV), as there were zero instances where the MBT Module reported a positive result that was contradicted by the reference PCR.

Suggestions:

1- Material and Methods section:

Your results show a very low sensitivity (15.5%), which is substantially lower than previously published metrics, which generally report sensitivities ranging from 42% to 85%. Was this dataset selected randomly? I believe the percentage of other Enterobacterales than *K. pneumoniae* could be potentially biasing such a low sensitivity. I see specifically an overrepresentation of species like *Citrobacter* and an underrepresentation of *Klebsiella*. In this sense, I suggest including information that supports the notion that this species' distribution aligns with your routine findings. It seems the number of *C. freundii* KPC - positive is too high. I would recommend reproducing in the study the approximate percentage of isolates you find in your routine results. Otherwise, the test performance could be severely affected.

The dataset was selected to follow the overall trend of KPC isolates from our institution. In the discussion section, we have elaborated on the point above (line 263-266), recognizing that the use of non-*Klebsiella* Enterobacterales may have reduced sensitivity. Between 2022-2023, the laboratory isolated 42 KPC-producing Enterobacterales, of which 19 were *K. pneumoniae* (45% of isolates) and 6 were *C. freundii* complex (14% of isolates). In the study dataset, there are 18 *C. freundii* complex isolates (out of 93, ~19%), which was comparable to our hospital distribution.

2- Results sections:

It would be very interesting to see the results in Table 3 present the performance of the proposed MALDI test for KPC-2 and KPC-3, separately, since the authors mention that the samples were selected based on previous sequence results.

KPC types were only available for the AR Bank isolates of which 5 were KPC-2 and 14 were KPC-3. As the MALDI-ToF only identified two AR Bank isolates (out of 19 known KPC positive isolates) and both were KPC-2, there is no additional information in this regard to

include in Table 3.

3- Discussion section:

The authors should discuss further whether this low sensitivity is due to geographical validation or sample biases, as this study is the first to use Canadian clinical isolates for this evaluation. The high prevalence of p019-negative KPC strains, inferred by the 15.5% sensitivity, suggests the local dominance of highly streamlined and potentially more promiscuous plasmids. Such plasmids, optimized for horizontal gene transfer, may selectively maintain the resistance gene (blaKPC), while jettisoning accessory elements like p019. Therefore, the low MBT sensitivity could function as an indirect indicator of a highly evolved, geographically adapted, and clinically resilient local plasmid reservoir. You should include this discussion in your manuscript.

We have added a section to the discussion to elaborate on this point further – citing the article by Gato et al. (2021) who observed wide geographic variation in *p019* amongst KPC *K. pneumoniae* isolates as well as the absence of *p019* from common plasmids (line 275-277). We have also noted that the low sensitivity in our study may be due to both our wide selection of Enterobacterales isolates and geographic variability.

4- Conclusion section:

The conclusion must unequivocally reiterate that 100% specificity and 100% PPV are the essential outcomes. This characteristic supports the MBT Module solely as a high-speed "rule-in" tool for initiating IPC and therapeutic triage earlier than current methods.

We have further stated in our conclusion that based on the findings of this study, “the MBT Module is a potentially useful adjunctive "rule-in" tool at the time of MALDI-ToF identification, prior to the availability of conventional susceptibility testing” (line 307-308).

General suggestions:

It would be exciting and valuable to see the results for the mature KPC protein compared to the p019 protein in your dataset.

We have not done a comparison between detection of the mature KPC protein and the p019 protein via MALDI-ToF, but have noted in the discussion that this could be an area of future direction (line 258-259).

Reviewer #2 (Comments for the Author):

This study evaluates MBT module to identify KPC harboring organisms. While the data of this study is not promising, the negative data may have utility to educate other clinical laboratorians to steer away from using this assay. Overall, it is a sound validation but could use some improvements in displaying data. Furthermore, their conclusions do not match the data presented.

Major revisions

Line 68. Consider removing figure 1 as this is reused data and graphics from a different source

Thank you for this suggestion. This data was collected from several reports published by the Provincial Infection Control Network of British Columbia. The purpose of this figure was to pull all the KPC data from these reports and present the information in a single figure, which to our knowledge, has not previously been done. Presentation of these data is important to contextualize KPC in British Columbia given the wide geographic variation that has previously been described in the literature. As such, we have decided to keep the figure in the manuscript as it provides a succinct representation of the key findings.

Line 99. Meropenem-vaborbactam is the drug of choice for KPC and the other drugs mentioned may not work if it is another carbapenemase. Consider revising since this study is only for KPC.

This sentence is informed by the IDSA 2024 Guidance on the Treatment of Antimicrobial Resistant Gram-Negative Infections. In the section on management of KPC infections outside the urinary tract, meropenem-vaborbactam, ceftazidime-avibactam, imipenem-cilastatin-relebactam are mentioned as preferred options with ceftiderocol acting as an alternative. The guideline panel includes that they slightly favor meropenem-vaborbactam, but the other antibiotics listed can still be used. As such, we will keep all four drugs in the manuscript to reflect the current guidelines.

Line 104. Several studies have looked at KPC detection with MALDI. Please cite here.

We have added citations as requested (line 108).

Line 121. In table 1, consider organizing members of the *E. cloacae* complex and *K. pneumoniae* complex together as inaccuracies speciating can occur via MALDI.

As per the reviewers recommendation, we have organized members of the *E. cloacae* complex and *K. pneumoniae* complex together and we have corrected Table 1 and 3.

Line 168. I would add negative predictive value to best illustrate impact from false negatives.

We have added the negative predictive value to the methods (line 181-183) and results (line 191-192) to illustrate the impact from false negatives.

Line 172. Table 3 needs more detail of the results or consider deleting. To make table 3 better, I would add total number of isolates for each species, percent with KPC, sensitivity with numbers, and specificity without numbers.

We have modified Table 3 as suggested by the reviewer, adding total number of isolates for each species and the sensitivity. We have not included the percent with KPC or the specificity without numbers as this can be found within the body of the text.

Line 182. Although previous studies show poor results, why are the results of this study even worse? I would like to see a possible explanation.

In the discussion, we have noted that the low sensitivity in our study may be due to both our wide selection of Enterobacterales isolates and geographic variability (line 275-288). As this was a validation, we selected a wider distribution of Enterobacterales isolates to better reflect our historic clinical laboratory findings. To substantiate our hypothesis on geographic variability, we cited the work by Gato et al. (2021) who observed the geographic variation of *p019* carriage in European KPC positive *K. pneumoniae* isolates (line 275-277).

Line 228. I disagree with the conclusions as an adjunctive tool. With 100% specificity all we know is that if it is right it is definitely right but if it is negative we should still be concerned. I do not think this would help stewardship efforts at all. What would be the proposed implementation. I would like to see conclusions regarding the actual time saved with these poor results. Overall, I find the author's conclusions do not match what the data says.

Although the sensitivity of this assay is admittedly poor; as the reviewer points out, a positive result can be relied upon to be positive. Since this assay runs in parallel to standard MALDI testing, no additional testing steps are needed for the lab's workflow and positive results appear on the MALDI report. (No result for this assay is recorded for negative results). As such, positive results can be called to the most responsible physician to suggest antibiotic coverage active against KPC and to the stewardship team to implement contact precautions for a CPE. These measures can be undertaken at the time of organism identification rather than ~18-24h later when susceptibility information would be available. Regardless of whether the MBT module result is positive or negative, organisms would continue to be tested for routine susceptibility as per institution protocol. Only in the case of a positive result would anything be expected to change, and in those small number of cases, antibiotic switch and IPAC control measures could be implemented more quickly.

Our proposed implementation is described in Figure 2. In our hospital laboratory, "KPC detection and clinical notification occurs only after susceptibility testing and specific carbapenemase testing with CARBA 5 and/or CPE PCR. The MBT Module enables earlier detection of KPC at the time of bacterial identification, allowing for timely clinical notification (about a day earlier than the current pathway)."

Was any investigation into the presence of *p019* in these isolates or variant of KPC? Phenotypic comparison would also be helpful via mCIM or Carba5.

No investigation was completed to examine the presence of *p019* in these isolates. Phenotypic methods were used in the initial work-up of clinical and surveillance isolates (line 116). Clinical isolates were tested with Carba5 and surveillance isolates were screened on CHROMagar™ mSuperCARBA media.

Re: Spectrum02511-25R1 (**Evaluation of the Bruker MALDI Biotyper® (MBT) Subtyping Module for detection of *Klebsiella pneumoniae* Carbapenemase (KPC) in Enterobacterales in a Canadian clinical microbiology laboratory**)

Dear Dr. Marthe Charles:

Your manuscript has been accepted, and I am forwarding it to the ASM production staff for publication. Your paper will first be checked to make sure all elements meet the technical requirements. ASM staff will contact you if anything needs to be revised before copyediting and production can begin. Otherwise, you will be notified when your proofs are ready to be viewed.

Sincerely,
Paul Luethy
Editor
Microbiology Spectrum